# Sex-Dependent Regulation of Placental Oleic Acid and Palmitic Acid Metabolism by Maternal Glycemia and Associations with Birthweight

**DOI:** 10.3390/ijms23158685

**Published:** 2022-08-04

**Authors:** Oliver C. Watkins, Hannah E. J. Yong, Tania Ken Lin Mah, Victoria K. B. Cracknell-Hazra, Reshma Appukuttan Pillai, Preben Selvam, Neha Sharma, Amaury Cazenave-Gassiot, Anne K. Bendt, Keith M. Godfrey, Rohan M. Lewis, Markus R. Wenk, Shiao-Yng Chan

**Affiliations:** 1Department of Obstetrics and Gynaecology, Yong Loo Lin School of Medicine, National University of Singapore, Singapore 119077, Singapore; 2Singapore Institute for Clinical Sciences, Agency for Science, Technology and Research, Singapore 117609, Singapore; 3NIHR Southampton Biomedical Research Centre, University of Southampton and University Hospital Southampton NHS Foundation Trust, Southampton SO17 1BJ, UK; 4Department of Biochemistry and Precision Medicine TRP, Yong Loo Lin School of Medicine, National University of Singapore, Singapore 119077, Singapore; 5Singapore Lipidomics Incubator, Life Sciences Institute, National University of Singapore, Singapore 119077, Singapore; 6MRC Lifecourse Epidemiology Centre, University of Southampton, Southampton SO17 1BJ, UK; 7Institute of Developmental Sciences, Faculty of Medicine, University of Southampton, Southampton SO17 1BJ, UK

**Keywords:** placenta, fatty acid, lipids, β-oxidation, fetal sex, diabetes, fetal growth

## Abstract

Pregnancy complications such as maternal hyperglycemia increase perinatal mortality and morbidity, but risks are higher in males than in females. We hypothesized that fetal sex-dependent differences in placental palmitic-acid (PA) and oleic-acid (OA) metabolism influence such risks. Placental explants (*n* = 22) were incubated with isotope-labeled fatty acids (^13^C-PA or ^13^C-OA) for 24 or 48 h and the production of forty-seven ^13^C-PA lipids and thirty-seven ^13^C-OA lipids quantified by LCMS. Linear regression was used to investigate associations between maternal glycemia, BMI and fetal sex with ^13^C lipids, and between ^13^C lipids and birthweight centile. Placental explants from females showed greater incorporation of ^13^C-OA and ^13^C-PA into almost all lipids compared to males. Fetal sex also influenced relationships with maternal glycemia, with many ^13^C-OA and ^13^C-PA acylcarnitines, ^13^C-PA-diacylglycerols and ^13^C-PA phospholipids positively associated with glycemia in females but not in males. In contrast, several ^13^C-OA triacylglycerols and ^13^C-OA phospholipids were negatively associated with glycemia in males but not in females. Birthweight centile in females was positively associated with six ^13^C-PA and three ^13^C-OA lipids (mainly acylcarnitines) and was negatively associated with eight ^13^C-OA lipids, while males showed few associations. Fetal sex thus influences placental lipid metabolism and could be a key modulator of the impact of maternal metabolic health on perinatal outcomes, potentially contributing toward sex-specific adaptions in which females prioritize survival.

## 1. Introduction

Perinatal mortality and morbidity are higher in males than in females, but mechanisms underlying such sex differences remain inadequately understood. Males are generally born heavier than females, while females have greater fat mass and are more likely to survive suboptimal in utero conditions such as uteroplacental insufficiency and maternal hyperglycemia [1,2,3,4]. It has been suggested that female fetuses are more adaptable to in utero adversity, while male fetuses prioritize a growth strategy at the expense of survival [5]. Understanding the specific pathways involved in these sex-dependent differences could lead to novel strategies to improve offspring outcomes. It is postulated that sex-dependent differences in placental function and metabolism may partly account for these differences.

### 1.1. Palmitic Acid and Oleic Acid in Pregnancy

Palmitic acid (PA, 16:0) and oleic acid (OA, 18:1) constitute 60% of fatty acids in lipids present in the maternal plasma by the end of pregnancy, and in both placenta and the fetus, these fatty acids act as fuel and as building blocks for the synthesis of structural and signaling lipids [6,7,8]. Both PA and OA are derived from diet and endogenous synthesis; thus, the fetus relies on both transplacental transfer and endogenous fetal–placental synthesis [9]. Therefore, sex-dependent variations in these processes may impact fetal growth and development [9].

Excess PA is often negatively depicted for its potential involvement in the pathophysiology of adult chronic diseases such as cardiovascular disorders and diabetes, and excess intracellular un-esterified PA may induce lipotoxic effects [7,8], including in the placenta [10]. However, PA is also an essential fuel source, is a constituent of many signaling lipid molecules, is needed for the regulation of enzymatic activity though palmitoylation [7], and is involved in the production of pro-inflammatory cytokines [11], which are important in regulating pregnancy health and parturition. Conversely, OA is often positively depicted, showing no pro-inflammatory effects and appearing to counter many damaging effects of excess PA [8,11,12]. For example, OA appears to protect trophoblasts from palmitate-induced death in vitro, possibly by inducing PA to be safely compartmentalized within lipid droplets [10,13,14]. OA also increases placental amino acid transport and phosphorylation of ERK, mTOR, S6 kinase 1, and rpS6 [11,12] to regulate placental function including fetal nutritional supply.

### 1.2. The Influence of Maternal Glycemia, BMI and Fetal Sex on Maternal and Placental Lipids

Higher maternal glycemia and BMI are associated with increased maternal plasma lipids, which may increase the risk of fetal macrosomia [9,15] despite regulation of transplacental lipid transfer by placental lipid transport and metabolism [16]. Independently of fatty acid supply, maternal glycemia and BMI may also separately influence placental lipid composition to affect pregnancy outcomes in a sex-dependent manner. Maternal obesity is associated with increased placental uptake of radio-labeled OA in females but decreased uptake in males [17]. However, most placental lipidomic studies to date have not considered sex, making it difficult to explore the extent to which sex-dependent differences in placental lipid metabolism might be implicated in pathophysiological processes.

### 1.3. Aims and Hypotheses

In this study, we aimed to measure the incorporation of ^13^C-PA and ^13^C-OA into lipids in cultured placental villus explants. We hypothesized that there are sex-dependent differences specifically in the placental processing of PA and OA, independent of maternal fatty acid supply and fetal utilization. We also hypothesized that maternal glycemia and BMI associate with placental PA and OA metabolism in a sex-dependent manner and that such sex-dependent alterations are linked with birthweight.

## 2. Results

### 2.1. Placental Production of Fresh ^13^C-PA- and ^13^C-OA-Containing Lipids Quantified in Explant Tissue and in Conditioned Media

Stable isotope-labeled ^13^C-PA and ^13^C-OA were taken up and metabolized by placental explants into stable isotope-labeled lipids as well as being funneled into other metabolic processes such as elongation (the addition of carbons to a fatty acid chain) or B-oxidation (the removal of carbons from fatty acid chains for use as fuel) (graphical abstract and Figure 1). Quantifying the amount of each individual ^13^C-PA or ^13^C-OA metabolite in placental explants by LCMS thus enabled the metabolic capacity of different placentas to be compared.

^13^C-PA and ^13^C-OA were incorporated into freshly produced stable-isotope-labeled glycerolipids (triacylglycerol (TG), diacylglycerol (DG)) and phospholipids (phosphatidylcholine (PC), phosphatidylethanolamine (PE), phosphatidylethanolamine-plasmalogen (PE-P), phosphatidylinositol (PI), lysophosphatidylcholine (LPC), lysophosphatidylethanolamine (LPE)) (Figure 1 and graphical abstract). ^13^C-PA was also incorporated into sphingolipids including ceramides (Cer) and sphingomyelins (SM). Both ^13^C-PA and ^13^C-OA were metabolized into acylcarnitines and acylcarnitine beta oxidation products, and ^13^C-PA was elongated into a ^13^C_16_ labeled acylcarnitine 18:0. Incorporation of one stable isotope labeled fatty acid was most common, but several di- or tri-labeled lipids were also detected. ^13^C-PA- and ^13^C-OA-labeled lysophospholipids and acylcarnitines were also found in the conditioned explant media at both 24 and 48 h. Since other labeled lipids were not found in media, even those abundant in placental explants, it seems likely that these lipids are specifically secreted/exported by the placental explants.

The proportion of phospholipids that contained a labeled ^13^C (amount of ^13^C lipid/amount of ^13^C lipid plus matching endogenous ^12^C lipid) was much lower (12% for ^13^C-OA phospholipids as a class, <3% for ^13^C-PA phospholipids as a class) than the proportion of ^13^C- TGs, DGs and acylcarnitines (30–50% for ^13^C-OA lipids, 7–26% for ^13^C-PA lipids). This suggests that the turnover and replacement of glycerolipids and acylcarnitines is much faster than phospholipids, consistent with previous findings [18].

### 2.2. Explants from Female Placentas Incorporate More ^13^C Lipids than Those from Male Placentas

Placental explants from females contained more of almost all ^13^C-OA lipids at 24 and 48 h, and ^13^C-PA lipids at 24 h of incubation (Figure 2A,B). That this sex difference was statistically significant for most lipids in most lipid classes suggests that sex-dependent events have a large impact on upstream processes such as fatty acid uptake or activation early during culture, leading to increased synthesis of diverse lipid classes. That an upstream process is involved is also supported by our finding that sex differences in the ^13^C-PA lipids are more apparent at 24 h than 48 h, since uptake and other related processes will become less rate limiting over time.

Conditioned media incubated with ^13^C-OA and explants from female placentas contained more ^13^C-OA LPC at 24 and 48 h, and more acylcarnitine 18:1 and 16:1 at 48 h compared with male cases. Meanwhile conditioned media from ^13^C-OA incubations had more ^13^C-PA LPC at 24 and 48 h, and more ^13^C-PA-acylcarnitine 16:0 at 24 h in female than in male cases. Further, the proportion of newly synthesized ^13^C lipids found in conditioned media compared to explants (amount in media/amount in media + amount in explant) was also higher in female cases, particularly for ^13^C-PA LPC, ^13^C-OA LPC, ^13^C-OA acyl carnitine 18:1 and ^13^C-OA acyl carnitine 16:1 (Figure 2C,D), suggesting that increased export occurs on top of the increased synthesis in females. Sex differences in lipid amount in placental explants and conditioned media remained similar after adjusting for maternal BMI or glycemia.

### 2.3. Placental Production of ^13^C Lipids Associated with Maternal Characteristics

We then examined relationships between amount of ^13^C lipid and maternal glycemia (fasting and 2 h post-load in a mid-gestation oral glucose tolerance test (OGTT)) or maternal BMI (Figure 3 and Figure 4). Of these measures, maternal 2 h glycemia (the time-point at which glycemia is most commonly elevated and fulfilling the criteria for gestational diabetes in our population [19,20]) showed the greatest number of associations with the amount of placental ^13^C-OA and ^13^C-PA lipids.

Among ^13^C-OA lipids, amounts of ^13^C-OA-TGs at 24 h were generally negatively associated with 2 h glycemia, which was significant for ten of them after adjusting for fetal sex (the combined analyses with males and females not shown, but lipids involved are indicated by *S in Figure 3C,D). In relation to glycemia, there were sex differences in its association with some phospholipids and acylcarnitines, including acylcarnitines found in conditioned media (significant sex*fasting-glycemia or sex*2 h-glycemia interactions are indicated by “►” in Figure 3). In sex-stratified analyses, female cases generally displayed a positive association for multiple ^13^C-OA-acylcarnitines with 2 h glycemia, while in males, more negative trends for TGs and the phospholipid ^13^C-OA-PE 36:2 were seen with increasing 2 h glycemia (Figure 3C), and positive associations for phospholipids (PE 34:1, PC 34:1, LPC 18:1) were observed with fasting glycemia (Figure 3B). Increasing divergence between males and females in the amount of these lipids were observed with increasing 2 h glycemia (Figure 3E). Overall, maternal BMI was positively associated with only ^13^C-OA acylcarnitine 18:1 and ^13^C-OA PC 34:1 after adjusting for sex, with no interactions observed between sex and BMI on ^13^C-OA lipids.

Among ^13^C-PA lipids, there was a general increase in the amount of ^13^C-PA-DGs (DG 32:0 (mono-labeled and di-labeled), and DG 36:3) and ^13^C-PA phospholipids (significant for the two most abundant phospholipids ^13^C-PA-PE 32:0 and PE 36:4) in association with increasing maternal 2 h glycemia at 48 h of culture (X* in Figure 4C,D). Further, there was a positive association with several ^13^C-PA derived acylcarnitines (^13^C-labeled-acylcarnitine 14:0 (explant) and 12:0 (explant and media) derived from beta-oxidized ^13^C-PA). Fasting glycemia also showed a general trend of positive associations with ^13^C-PA lipids, which were significant for DG 34:2 and TG 54:4 at 48 h. These associations remained similar after adjusting for fetal sex. However, significant interactions between sex and 2 h glycemia were observed for several ^13^C-PA lipids at 24 h. Sex-stratified regression models showed generally positive associations between ^13^C-PA lipids and 2 h glycemia in female, but not in male cases (Figure 4C,E). In females, significantly positive associations with 2 h glycemia were seen with the amount of four ^13^C-PA-derived acylcarnitines, two ^13^C-PA DGs, three ^13^C-PA phospholipids and one ^13^C-PA sphingolipid. Similar to ^13^C-OA lipids, these sex differences also showed increasing divergence with increasing 2 h glycemia (See selected scatter plots in Figure 4E). Maternal BMI was positively associated with only ^13^C-PA DG 32:0 (both mono and di-labeled) after adjusting for sex. Even though there were no significant interactions between sex and BMI, increases in the amount of ^13^C-PA DGs and ^13^C-PA-acylcarnitine with increasing BMI were more apparent in females.

### 2.4. Birthweight Centile Is Associated with Placental ^13^C-OA Lipids and ^13^C-PA Lipids in Female Cases

We then investigated whether variations in the amount of explant or conditioned media ^13^C-OA or ^13^C-PA lipids could be associated with birthweight centile (standardized for gestational age using local reference). In a combined model including males and females, there was no association between any explant ^13^C-PA or ^13^C-OA lipids and birthweight centile at either 24 or 48 h. Adjusting for fetal sex, glycemia or BMI did not alter these results. However, two acylcarnitine species exported into the media at 48 h (^13^C-OA-acylcarnitine-14:1-CM and ^13^C-PA-acylcarnitine-12:0-CM) were positively associated with birthweight centile. Since significant interactions were found between sex and several ^13^C-OA or ^13^C-PA lipids on birthweight centile, sex-stratified analyses were conducted. Among female cases, generally negative associations were observed for ^13^C-OA-TGs at 24 h and positive associations for ^13^C-OA and ^13^C-PA acylcarnitines at 48 h. In contrast, few lipid species showed associations with birthweight in male cases.

Several other lipids also showed sex-divergent associations with birthweight (represented by ► in Figure 5 where there was significant interaction sex*amount of lipid on birthweight centile): ^13^C-OA-LPE showed a positive trend in males but negative in females; ^13^C-PA-SM 34:2, ^13^C-PA-DGs, and ^13^C-PA Cer-d18:2/16:0 showed negative trends in males but positive in females. This suggests that sex-dependent placental lipid metabolism may also play a part in explaining sex-dependent perinatal outcomes such as birthweight. Of note, the positive association coefficient between ^13^C-PA lipids and birthweight centile in female cases were particularly high for ^13^C_16_-PA acylcarnitine 16:0 (the major ^13^C-PA-derived acylcarnitine; estimate (95% CI) 53 (41–65) birthweight percentile units/(Log-2 pmol/mg)). This is equivalent to 40 birthweight percentile units per SD increase in ^13^C_16_-PA acylcarnitine 16:0.

Many of the ^13^C-OA and ^13^C-PA acylcarnitine species associated with birthweight centile in females were also positively associated with maternal glycemia (^13^C-OA-derived acylcarnitine 14:1-CM, 16:1 and ^13^C-PA-derived acylcarnitine 12:0, 14:0, 16:0) or BMI (^13^C-OA-derived acylcarnitine 14:1, 14:1-CM, 16:1 and ^13^C-PA-derived acylcarnitine 16:0). Thus, the increase in placental acylcarnitine synthesis appears to reflect key events by which higher maternal glycemia or BMI may lead to greater fetal growth.

### 2.5. The Relationship between mRNA Expression of Placental Fatty Acid Uptake Genes with Sex and Placental OA and PA Lipid Metabolism

Since most ^13^C-PA and ^13^C-OA lipids were higher in females than males, we hypothesized that these variations in placental lipid abundance could be due to sex differences in the expression of genes responsible for fatty acid uptake. However, when we investigated the placental mRNA abundance of fatty acid transport genes (*SLC27A 1,2,3,4,6* and *CD36*), fatty acid binding proteins (*FABP 3,4,5*) or acyl CoA synthetases (*ACSL 1,3,4,5,6*) we found no significant difference in the relative expression levels in males compared with females (Figure 6).

However, our results suggest that the relative mRNA expression of these fatty acid uptake genes plays a significant role in regulating the fresh synthesis of specific placental lipids, with some sex differences. The relative mRNA expression of placental fatty acid uptake genes was generally positively associated with increased ^13^C-PA lipids at 48 h (Figure 7, *X), especially for *ACSL3* (two DGs), *ACSL4* (26 lipids), *SLC27A2* (14 lipids), and *SLC27A6* (three lipids). In sex-stratified analyses, the abundance of many ^13^C-PA lipids was positively associated with *ACSL3*, *ACSL4*, *ACSL6*, *FABP5*, *SL27A1*, *SL27A2* and *SL27A3* mRNA in females, while positive associations were only seen with *ACSL4* mRNA in males. In contrast, mRNA expression of placental fatty acid uptake genes was generally not positively associated with ^13^C-OA lipids apart from *SLC27A2* and ^13^C-OA TG 58:9 (Figure 8, *X). Instead, there were numerous significant negative associations seen for *CD36* (PC 38:4), *ACSL1* (four acylcarnitines), *FABP3* (six lipids), *SL27A1* (three lipids), *SLC27A3* (PC 38:4). In sex-stratified analyses, *ACSL3* mRNA was positively associated with ^13^C-OA acylcarnitines (three lipids) while *SLC27A2* mRNA was positively associated with six ^13^C-OA TGs in females. Meanwhile in males, negative associations were observed between transcripts of *FABP3*, *FABP4*, and *SLC27A2* with ^13^C-OA lipids.

## 3. Discussion

### 3.1. Main Findings

The metabolism of the two most abundant fatty acids, OA and PA, by the placenta is strongly sex-dependent, and alters in response to variations in the in utero environment imposed by increasing maternal glycemia and BMI. In vitro, female placenta produced more ^13^C-OA- and ^13^C-PA-labeled lipids from every measured lipid class, suggesting that female placenta have generally greater capacity for fatty acid uptake and production. However, female placenta did not show increased mRNA expression of any of the tested placental fatty acid transporters or fatty acid CoA activation enzymes (a necessary step preceding lipid synthesis). Thus, increases in lipid production in females instead likely involve regulation at a post-transcriptional level such as increases in protein expression, activity or other yet-to-be-elucidated processes. Sex-dependent differences in OA and PA processing may lead to greater placental fat storage and fetal supply in females, consistent with findings of greater percentage body fat in female neonates than males [21]. It also supports the idea that females are more protected against sub-optimal nutritional supply such as in conditions of uteroplacental insufficiency, as well as in gestational diabetes and obesity, as reflected by lower perinatal morbidity and mortality compared with their male counterparts [5]. In females, but not males, we found positive associations between placental acylcarnitine metabolism and maternal glycemia, BMI, and birthweight. We postulate that regulation of placental beta-oxidation plays an important role in enabling female fetuses to better adapt to challenging in utero conditions, and they therefore display lower perinatal morbidity and mortality.

### 3.2. Sex Dependent Differences in Placental Lipid Metabolism

A previous ex vivo study had suggested that female placenta contained more endogenous un-esterified OA, OA-TGs, OA-acylcarnitines and more acylcarnitines of OA-beta-oxidation products than males [22]. Our in vitro findings of increased production of almost all OA and PA lipids in the placental explants of females compared to males suggest that such differences are a direct result of sex-dependent differences in placental lipid processing, rather than differences in maternal fatty acid supply or in fetal utilization. Such differences in placental lipid processing also likely explain the sex difference in potential lipid supply to the fetus, where venous cord blood (which contains nutritional supplies and signals picked up from the placenta for transport to the fetus) from females contains more un-esterified fatty acids (including saturated and mono-saturated fatty acids), phospholipids and glycerolipids than males [23].

Many tissues demonstrate sex differences in fatty acid uptake, postulated to be due to differences in the expression and activity of genes involved in fatty acid uptake. For example, female mouse heart and rat liver show greater fatty acid uptake compared to males [24,25,26,27,28,29], and females show greater rat liver protein expression of FABP [26,28,29] and greater *CD36* mRNA expression in several human and rat tissues [30,31,32]. Our findings suggest that in the placenta, sex differences in lipid abundance are not the result of simple increases in the mRNA expression of *ACSL*, *FABP* or *SLC27* gene families. Instead, differences are likely due to augmentation in protein expression or activity of these genes, alterations in the operation of other sex-specific fatty acid transport mechanisms and other downstream lipid metabolic processes, or differences in the regulatory importance of fatty acid uptake compared to other lipid synthetic or catabolic processes. This is evidenced by the positive association between ^13^C-PA lipids with *ACSL3*, *ACSL6*, *FABP5* and *SLC27A2*, and ^13^C-OA lipids with *ACSL3* and *SLC27A2* in females, but a negative association between ^13^C-OA lipids with *FABP3* and *FABP4* in males.

Our own PA results and many other studies suggest that FABP enhances transport of saturated fatty acids into cells, and this is thought to be because FABPs facilitate the transfer of fatty acids across aqueous cellular regions [33]. However, FABP can also assist in fatty acid export [34,35] as well as in binding to and thus regulating the activity of signaling molecules such as hormones, lipids and eicosanoids [36,37]. Our findings of negative associations between ^13^C-OA lipids with *FABP3* and *FABP4* in males may suggest that the aforementioned non-fatty-acid-uptake processes may play an important role. This would be consistent with previous reports that increasing the expression of *FABP2* (*I-FABP*) in H141 cells decreased ^3^H-oleic acid uptake and its incorporations into phospholipids [38], suggesting net efflux, while *FABP4* null mice show both increased OA uptake and increased OA efflux [39].

### 3.3. The Influence of Maternal Glycemia and BMI on Placental OA and PA Lipid Production

In the human in vivo study where ^13^C-PA was administered alongside other ^13^C-fatty acids, GDM was associated with increased total placental ^13^C-PA lipid [40], consistent with our finding of generally positive associations between the amount of placental ^13^C-PA lipids with 2 h glycemia. The levels of endogenous placental non-esterified palmitic acid are also increased in GDM compared to controls [41]. Obesity reportedly had no influence on total ^13^C-OA placental lipid production in vitro [42] and GDM did not associate with any change in ^13^C-OA placental lipids in vivo [40], consistent with our own results showing no association between BMI or glycemia and ^13^C-OA placental lipids when sex was not considered.

While most existing literature does not report how glycemia might influence placental lipids in males and females separately, there are a few reports on sex differences in the influence of maternal BMI on placental OA lipids. One study reported a positive association between obesity and placental uptake of radio-labeled OA in females but a negative association in males [17]; placental lipid production was not quantified. Although we found that BMI was positively associated with ^13^C-OA and ^13^C-PA acylcarnitines in females, it seems unlikely that a general increase in fatty acid uptake would only increase acylcarnitines and not other lipid classes. Furthermore, we found that ^13^C-OA LPC, LPE and PC 34:1 positively associated with BMI in male cases, suggesting that the sex-dependent effects also demonstrate lipid-class specificity and predominantly occur in activities downstream from the fatty acid uptake process.

Greater serum LPC species have previously been reported to play a more important role in decreasing insulin sensitivity in adult males compared to adult females (2). Fatty acids meanwhile are bound to carnitine to form acyl-carnitines to facilitate their transport across the inner mitochondrial membranes, enabling beta oxidation and the utilization of fatty acids for energy [43]. Beta oxidation, similar to other alternative fuel pathways including glycolysis, polyol and pentose-phosphate pathways, ketogenesis and ketone oxidation, is important in the placenta, particularly as it is vulnerable to hypoxia exposure (e.g., with reduced maternal blood flow to the placental bed in situations such as regular uterine contractions in parturition, circulatory diversions to other maternal organs, maternal aortocaval compression) or to situations where the TCA cycle cannot easily be completed [44,45,46,47,48]. Acylcarnitines also act as signaling molecules and as sources of activated fatty acid for lipid remodeling and protein palmitoylation [49,50,51]. Thus, our findings of increased levels of acylcarnitines with increasing maternal 2 h glycemia and BMI in females, but not males, may represent a sex-specific adaptive strategy that promotes survival of female offspring in the face of in utero adversity. Increased beta oxidation could also reduce placental free-unesterified PA, which has lipotoxic effects in trophoblasts [10,11]. Increased beta oxidation could provide an alternative fuel source for the fetus if glucose utilization is impaired as observed with insulin resistance, hyperglycemia and obesity in adults [52,53,54].

Our findings of increased OA-acylcarnitines in explants with increasing BMI in females contrast with the reduced endogenous OA-related acyl-carnitines (18:1, 16:1, 14:1) and free carnitine found in placental microvillous membrane (MVM) isolates [22] from obese white Caucasian, non-GDM women carrying female compared to male offspring. Aside from the obvious experimental and population differences, this may reflect reduced carnitine availability in obesity rather than altered placental OA metabolism.

Female-specific associations of ^13^C-PA DGs with glycemia (2 h) and BMI and between ^13^C-OA LPC and LPE with maternal BMI may also suggest that alterations in these important lipid signaling molecules [55,56,57,58,59] may also be part of a sex-specific adaptive strategy. Placental DGs are known to induce the release of human chorionic gonadotropin and placental lactogen [60,61,62] and stimulate the production of progesterone [63].

### 3.4. Placental Lipids and Birthweight

Our findings of sex-dependent associations between newly synthesized placental PA and OA lipids and birthweight centile suggest the involvement of placental lipid metabolism in the sex-specific regulation of fetal growth (Figure 9). Associations with birthweight were mainly observed in females, with positive associations seen for OA and PA acylcarnitines and a PA ceramide, and negative associations for OA phospholipids and OA-TGs.

Ceramide d18:2_16:0 is one of most abundant PA lipids in the placenta and was positively associated with birthweight in females but was negatively associated in males. A previous study had found increased ceramide in placenta from normal-weight fetuses compared to those with intra-uterine growth restriction (IUGR), consistent with our current findings in females [64]. Ceramides likely affect fetal growth by acting as signaling molecules, decreasing the expression of nutrient transporters [65,66,67], and regulating placental inflammation, insulin signaling, AKT activation, and amino acid transport [65,66,67,68,69,70,71]. Negative associations of birthweight centile with OA-TGs in female cases, meanwhile, might indicate reduced sequestration of OA into TG storage, making it more available for transfer to the fetus, especially since cord plasma OA has previously been linked with increased birthweight [72].

Umbilical cord blood acylcarnitines are associated with being small or large for gestational age relative to appropriately grown controls [54,73,74,75]; however, there have been no studies of the role of placental acylcarnitines. We found that conditioned media showed increased 12:0 and 14:1 acylcarnitines in association with birthweight in females. Placentally synthesized acylcarnitines are known to be transferred to the fetus [43,76,77] as metabolic fuel as a source of carnitine and as signaling molecules potentially regulating insulin action and adiposity [47,76,77]. We speculate that in growth restriction, placental export of acylcarnitines acts mainly as an alternative fuel source to promote fetal survival, but with maternal hyperglycemia and obesity, increased placental acylcarnitines may also act as signals to limit excessive growth and hence perinatal morbidity and mortality.

### 3.5. Strengths and Limitations

Our targeted LCMS method allowed us to separately interrogate the metabolism of almost 100 different stable isotope-labeled lipids, enabling us to explore the importance of many different lipidomic pathways. Our study was able to quantify not only ^13^C-PA and ^13^C-OA acylcarnitine but also the acylcarnitine derivatives of ^13^C-PA and ^13^C-OA after beta oxidation and elongation, enabling us to trace PA and OA through several cycles of beta-oxidation, which was not possible in previous studies. Quantification of ^13^C_16_-labeled acylcarnitine 18:0 further enabled us to trace the elongated PA into stearic acid (18:0), the first documented evidence of placental PA elongation despite the placenta being known to express several elongases [78]. We were also able to quantify the export of ^13^C-acylcarnitines and lysophospholipids into conditioned media. However, placental explant-type experiments cannot indicate whether this represents export to the mother or to the fetus. Stable isotope-labeled placental explant experiments are difficult to perform on large numbers of placentas, limiting our ability to explore the effects of multiple clinical factors simultaneously or to explore mediation effects. As only two cases were treated with insulin, we were unable to assess how insulin might impact placental OA and PA lipid metabolism.

## 4. Materials and Methods

### 4.1. Placental Collection

Placenta were collected with written informed consent from 22 women recruited at the National University Hospital, Singapore. Placentas were collected from non-smoking mothers who delivered by elective Caesarean section after 37 weeks’ gestation and who underwent a three-time-point 75 g oral glucose tolerance test (OGTT) at mid-gestation. Cases were matched for first trimester BMI and GDM (WHO 2013 criteria) to ensure a balance of cases across the sexes (Table 1).

Cases of known pre-existing diabetes mellitus or possible pre-existing diabetes diagnosed in pregnancy (defined by antenatal OGTT results of fasting glycemia ≥ 7.0 mmol/L or 2 h glycemia ≥ 11.1 mmol/L) were excluded. Researchers were blinded to the clinical characteristics of the women and neonates involved until the completion of all explant culture experiments. Of the women with GDM, nine were diet controlled and only two were insulin treated.

### 4.2. Placenta Collection and Placental Explant Culture

Villous placental explants were cultured as previously described [79]. Briefly, five pieces of villous placental tissue were biopsied from five different sites across the placenta. These biopsies were then cut into small explants (approximately 3 mm^2^) and added to each experimental well, with one explant coming from each different placental biopsy. All placental processing was finished within two hours of delivery. Explants were cultured in 12-well plates (NunclonTM Delta surface, Thermofisher, Waltham, MA, USA) in serum-free CMRL media (1.8 mL, GIBCO 1066-L Glutamine, Thermofisher, Waltham, MA, USA) containing 5 mM glucose, P5030, Biowest, Nuaillé, France) with 1.5% BSA (HI Clone fraction V, Culture grade, pH 7.00 lyophilized powder, Cytiva, Marlborough, MA, USA) and either ^13^C_16_ PA (100 µM) or ^13^C_18_ OA (100 µM). Stable isotope-labeled fatty acids (>99 atom % ^13^C, >99% (CP)) were purchased from Sigma Aldrich (Saint Louis, MO, USA). Triplicate wells were run for each condition. Levels of PA and OA in maternal blood are highly variable, and those available for placental uptake in vivo during pregnancy are unknown since these fatty acids also originate from esterified plasma lipids. We therefore chose to add 100 µM to represent levels around those of maternal blood and cord blood [80,81], a concentration that was high enough to enable quantification of a range of ^13^C-labeled placental lipids in both placental explants and conditioned media.

Explants were incubated at 37 °C in a humidified atmosphere of 5% CO_2_/air for 24 or 48 h. Explants were then harvested with triplicates combined in a pre-weighed Omnitube (Waltham, MA, USA). Explants were washed with PBS (1 mL) and excess PBS removed after centrifugation. Explant media from triplicate wells (800 µL from each) were combined into an Eppendorf tube, allowed to settle for 5 min then clear supernatant (1 mL) transferred into a fresh Eppendorf. Samples were stored at −80 °C until lipid extraction.

### 4.3. RNA Extraction, cDNA Synthesis and Real-Time Quantitative Polymerase Chain Reaction (RT-qPCR)

Total RNA from frozen placental biopsies was extracted using phenol–chloroform method and further purified using the RNeasy Mini Kit (Qiagen, Hilden, Germany) following manufacturer’s instructions. RNA concentrations and quality were then determined by using a Nanodrop ND-1000 Spectrophotometer (Thermo Fisher Scientific, Waltham, MA, USA). Placental RNA was reverse transcribed to cDNA with Superscript III reverse transcriptase (Thermo Fisher Scientific, Waltham, MA, USA) according to the manufacturer’s protocol. To determine mRNA expression of lipid genes, RT-qPCR was performed with 5 ng cDNA in duplicate 10 µL reactions using TaqMan Fast Advanced Master Mix (Thermo Fisher Scientific, Waltham, MA, USA) on the Applied Biosystems 7500 Fast Real-Time PCR System (Thermo Fisher Scientific, Waltham, MA, USA) under the fast settings (95 °C for 20 s, followed by 45 cycles of 95 °C for 3 s and 60 °C for 30 s). Inventoried FAM-labeled TaqMan probes (Thermo Fisher Scientific, Waltham, MA, USA) were used for 3 housekeeping genes—*CYC1* (cytochrome C1, Hs00357718_m1), *SDHA* (succinate dehydrogenase complex, subunit A, Hs00188166_m1) and *TBP* (TATA-box binding protein, Hs00427620_m1); 5 ACSL family genes—*ACSL1* (Hs00242530_m1), *ACSL3* (Hs00244853_m1), *ACSL4* (Hs00244871_m1), *ACSL5* (Hs01061754_m1) and *ACSL6* (Hs00922295_m1); 3 FABP family genes—*FABP3* (Hs00997360_m1), *FABP4* (Hs01086177_m1) and *FABP5* (Hs02339439_g1); 5 SLC27A family genes—*SLC27A1* (Hs01587911_m1), *SLC27A2* (Hs01113391_m1), *SLC27A3* (Hs00225680_m1), *SLC27A4* (Hs00192700_m1) and *SLC27A6* (Hs00204034_m1) as well as *CD36* (Hs00354519_m1). Relative expression of each gene was then calculated by the formula 2^(−ΔCT)^ and normalized to the geometric mean of expression of the three control genes [82].

### 4.4. Lipid Quantification by LCMS and Statistical Analysis

Lipids were extracted and analyzed by LCMS as described in Appendix A. Placental lipid content was expressed as pmol lipid in explant/mg dry weight of explant, and lipid concentration in conditioned media was expressed as pmol lipid in media/mg dry weight of explant. For statistical analysis, lipid amounts were log2 transformed (to achieve an approximately normal distribution). Linear regression was run for each lipid (outcome: lipid amount at 24 or 48 h) with each variable of interest (predictor: fetal sex, maternal BMI or fasting/post-load 2 h glycemia). Linear regression models were run in R (Vienna, Austria) version "Kick Things" with ‘tidyverse’ Version: 1.3.1 and ‘mediation’ Version: 4.5.0 packages. Where indicated, multiple linear regressions were then performed with mutual adjustments for these variables. Interactions between sex and each maternal factor on lipid amount were observed for multiple lipids; hence, stratified analyses by sex were also conducted for each lipid. Birthweight centile (customized for gestational age using local standards) was also analyzed as an outcome against each lipid (predictor) with interactions observed between sex and lipid amount, leading us to perform sex-stratified linear regression. For all tests, the Benjamini–Hochberg method was used to correct for multiple testing of lipids and time-points to minimize false discovery with statistical significance set at a two-sided alpha level of *p* < 0.05.

## 5. Conclusions

Our study suggests that placental lipid metabolism is dependent on fetal sex. Female placentas produce more PA and OA lipids, and the stage at which such sex-dependent regulation occurs appears to be downstream of the transcriptional regulation of placental fatty acid uptake genes. Furthermore, fetal sex influences the relationship between maternal glycemia and maternal BMI with placental PA and OA lipid metabolism, as well as between placental PA and OA lipid metabolism and birthweight, with associations mainly observed in females rather than in males. We postulate that sex differences in placental lipid processing likely contribute to sex differences in perinatal outcomes. The ability to adapt placental lipid processing to decrease vulnerability to adverse conditions in utero potentially contributes toward the overall lower perinatal morbidity and mortality risk in female offspring.

## Figures and Tables

**Figure 1 ijms-23-08685-f001:**
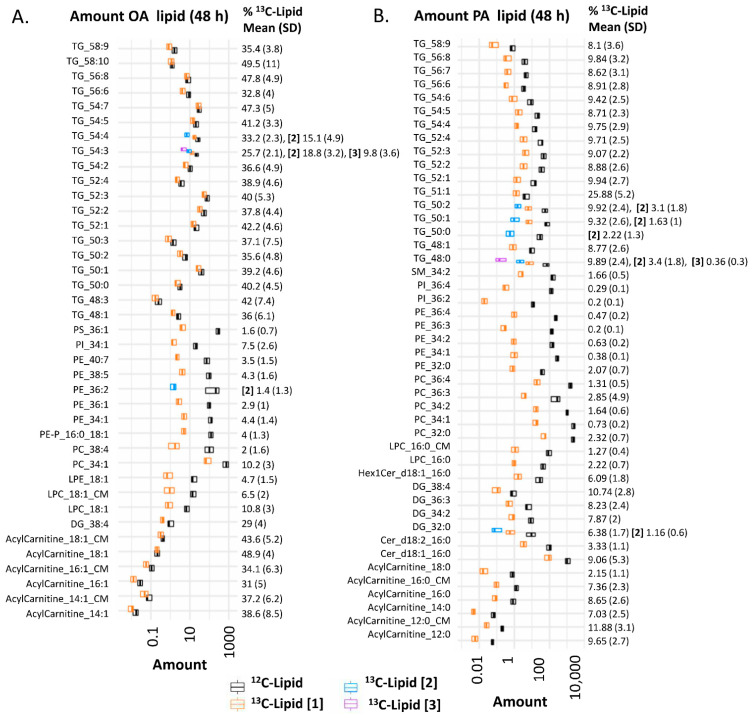
Amount of lipid in placental explants (*n* = 22) incubated in 100 µM ^13^C-OA (**A**), or 100 µM ^13^C-PA (**B**) for 48 h. Boxplots (median and interquartile range) show the amount (pmol lipid/mg of dry explant) of ^13^C fatty acid-labeled lipids (orange: mono-labeled [1], light blue: di-labeled, purple: tri-labeled [3]) and endogenous ^12^C lipids (black) in placental explants and in conditioned media (CM). x-axis is log10 scaled. Numbers to the right show percentage of each lipid labeled with ^13^C out of the total of that specific lipid (100*Amount ^13^C lipid/total amount of ^13^C + ^12^C of corresponding lipid) alongside standard deviation (SD). Bold square brackets show percentage of each particular lipid di- [2] or tri- [3] labeled with ^13^C-PA or ^13^C-OA. Abbreviations; DG: diacylglycerol, Cer: ceramide, Hex: hexose, LPC: lysophosphatidylcholine, LPE: lysophosphatidylethanolamine, OA: oleic acid, PA: palmitic acid, PC: phosphatidylcholine, PE-P: phosphatidylethanolamine-plasmalogen, TG: triacylglycerols.

**Figure 2 ijms-23-08685-f002:**
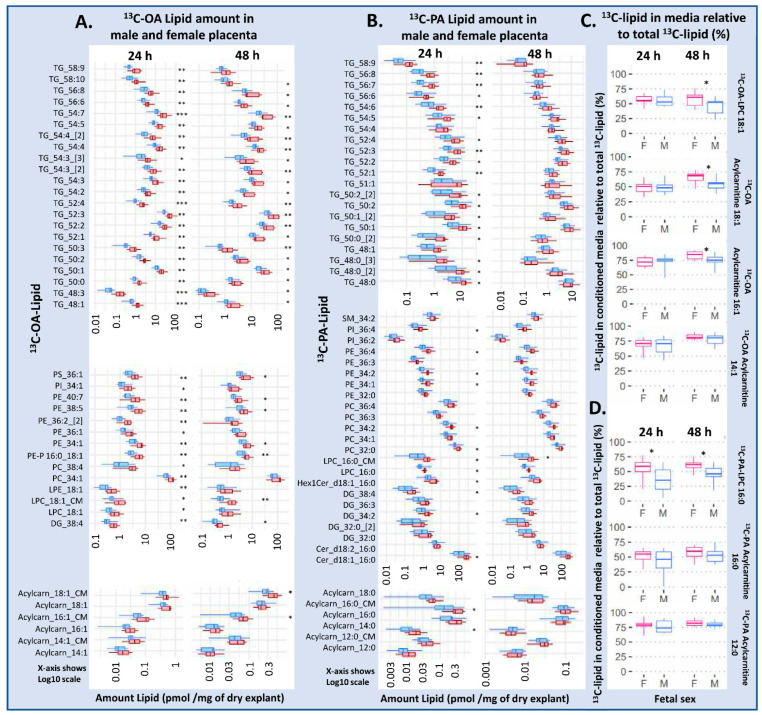
Differences in amount of lipid in placental explants from males (blue) and females (red) treated with 100 µM ^13^C-OA (**A**), or 100 µM ^13^C-PA (**B**) for 48 h. (**C**,**D**) The percentage of labeled lipid found in media relative to the total amount of the specific ^13^C lipid in the media and explant, in males and females. Boxes and whiskers represent median, interquartile range and minimum and maximum. X axis is log10 scaled. Statistically significant differences between sexes following Benjamini–Hochberg correction for multiple testing * *p* < 0.05, ** *p* < 0.01, *** *p* < 0.001. Abbreviations, Cer: ceramide, DG: diacylglycerol, Hex1Cer: hexosylceramide, LPC: lysophosphatidylcholine, LPE: lysophosphatidylethanolamine, OA: oleic acid, PA: palmitic acid, PC: phosphatidylcholine, PE-P: phosphatidylethanolamine-plasmalogen, TG: triacylglycerols. F: female, M: male.

**Figure 3 ijms-23-08685-f003:**
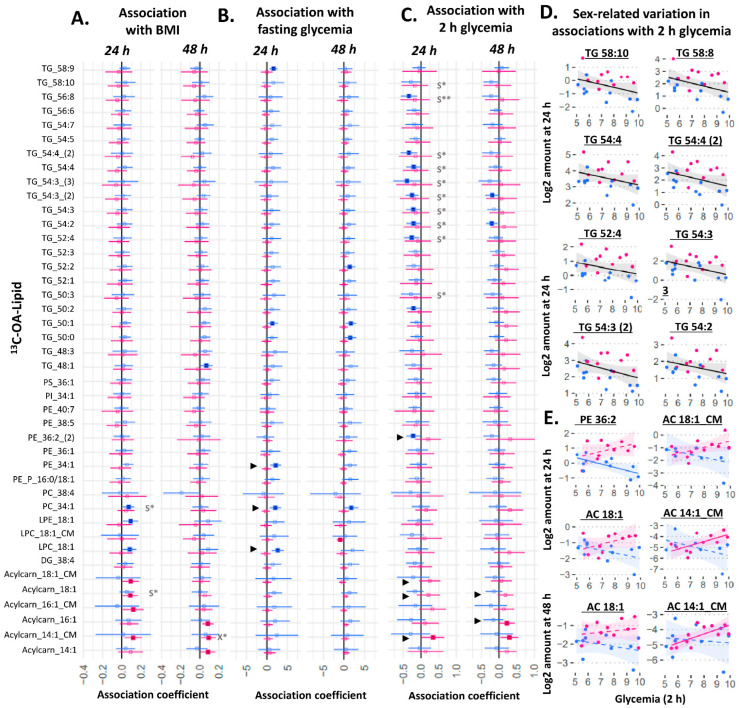
Sex-stratified associations between the amount of ^13^C-OA lipid (outcome) and maternal glycemia or maternal BMI (predictor). X* represents *p* < 0.05 in linear regression with males and females combined (not shown). S* represents *p* < 0.05, while S** represents *p* < 0.01 in linear regression adjusted for sex (not shown). ► represents significant *p* < 0.05 for the interaction between sex and maternal glycemia on lipid amount. Linear regression was then run separately for males and females. Forest plots (**A**–**C**) show association coefficient and 95% confidence intervals for males (blue) and females (red) with significance (*p* < 0.05) shown by dark-colored filled squares. Unadjusted scatter plots illustrating examples of the relationship between 2 h glycemia and amount of lipid for (**D**) lipids where males and females show similar relationships and for (**E**) lipids where males and females show different relationships. The Benjamini–Hochberg method was used to correct for multiple testing. Abbreviations: Acylcarn and AC: acylcarnitine, DG: diacylglycerol, LPC: lysophosphatidylcholine, LPE: lysophosphatidylethanolamine, OA: oleic acid, PA: palmitic acid, PC: phosphatidylcholine, PE-P: phosphatidylethanolamine-plasmalogen, TG: triacylglycerols.

**Figure 4 ijms-23-08685-f004:**
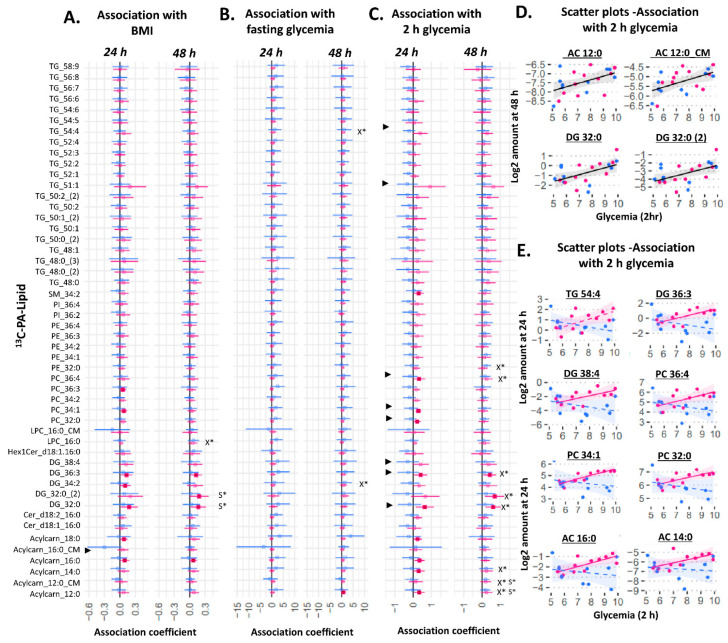
Sex-stratified associations between the amount of ^13^C-PA lipid (outcome) and maternal glycemia or maternal BMI (predictor). X* represents *p* < 0.05 in linear regression with males and females combined (not shown). S* represents *p* < 0.05 in linear regression adjusted for sex (not shown). ► represents significant *p* < 0.05 for the interaction between sex and maternal glycemia on amount of lipid. Linear regression was then run separately for males and females. Forest plots (**A**–**C**) show association coefficient and 95% confidence intervals for males (blue) and females (red) with significance (*p* < 0.05) shown by dark-colored filled squares. Unadjusted scatter plots illustrating the relationship between 2 h glycemia and amount of lipid for (**D**) lipids where males and females show similar relationships and for (**E**) lipids where males and females show different relationships. The Benjamini–Hochberg method was used to correct for multiple testing. Abbreviations: Acylcarn and AC: Acylcarnitine, DG: diacylglycerol, Cer: ceramide, Hex: hexose, LPC: lysophosphatidylcholine, LPE: lysophosphatidylethanolamine, OA: oleic acid, PA: palmitic acid, PC: phosphatidylcholine, PE-P: phosphatidylethanolamine-plasmalogen, TG: triacylglycerols.

**Figure 5 ijms-23-08685-f005:**
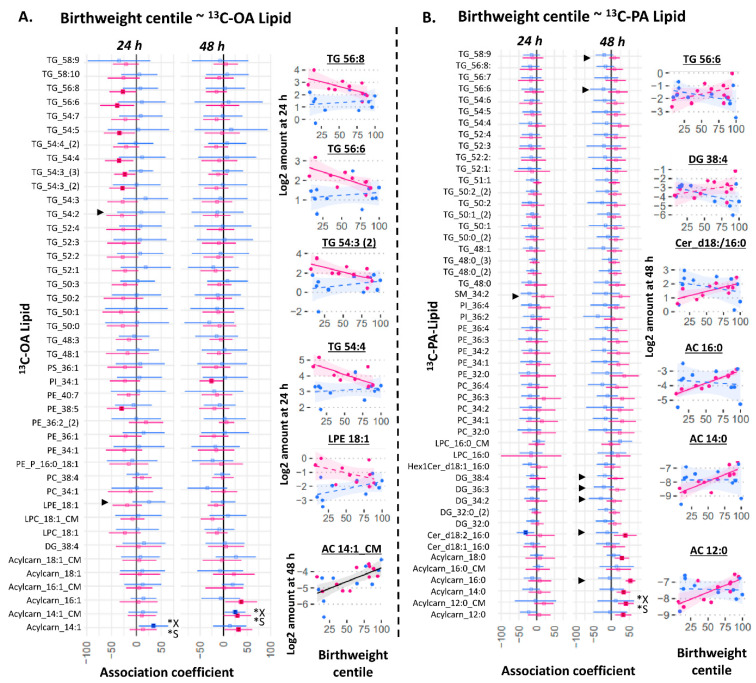
Sex-stratified associations between birthweight centile (outcome) with the amount of placental ^13^C-OA lipid (**A**) or ^13^C-PA lipid (**B**) after 24 or 48 h of culture. Birthweight centile was standardized for gestational age by local references. X* represents *p* < 0.05 in linear regression with males and females combined (not shown). S* represents *p* < 0.05 in linear regression adjusted for sex (not shown). ► represents significant *p* < 0.05 for the interaction between sex and amount of lipid on birthweight centile. Linear regression was then run separately for males and females (shown in figure). Forest plots (A,B) show association coefficient and 95% confidence intervals for males (blue) and females (red) with significance (*p* < 0.05) shown by dark-colored filled squares. Unadjusted scatter plots illustrating the relationship between birthweight centile and amount of lipid. The Benjamini–Hochberg method was used to correct for multiple testing. Abbreviations: Acylcarn and AC: Acylcarnitine, DG: diacylglycerol, Cer: ceramide, Hex: hexose, LPC: lysophosphatidylcholine, LPE: lysophosphatidylethanolamine, OA: oleic acid, PA: palmitic acid, PC: phosphatidylcholine, PE-P: phosphatidylethanolamine-plasmalogen, TG: triacylglycerols.

**Figure 6 ijms-23-08685-f006:**
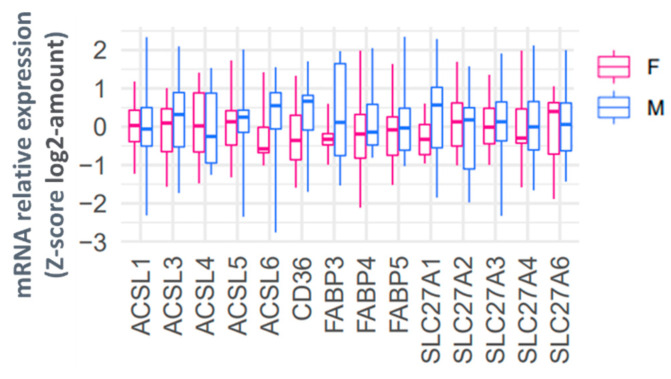
Relative placental mRNA expression of genes involved in fatty acid uptake in females (red) and males (blue). mRNA abundance is normalized to the geometric mean of three housekeeping genes and then log2 transformed and converted to a z-score. Boxes show median and interquartile range while whiskers show minimum and maximum values.

**Figure 7 ijms-23-08685-f007:**
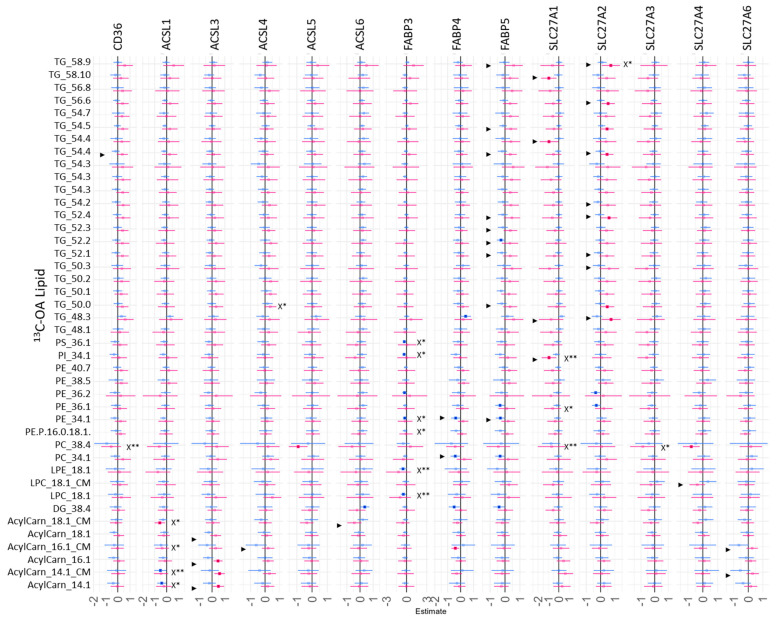
Sex-stratified associations between the amount of ^13^C-OA lipids (outcome) with relative mRNA abundance after 48 h of culture. Lipid amount was log2 transformed, mRNA abundance was log-2 transformed and z-scored. Linear regression with males and females combined (not shown) with significance *p* < 0.05 is indicated by X*, whilst significance *p* < 0.01 is indicated by X**. Interaction between sex*relative mRNA abundance on lipid amount were then investigated with significant interactions *p* < 0.05 denoted by “►”. Linear regression was then run separately for males and females. Forest plots show association coefficient and 95% confidence intervals for males (blue) and females (red) with significance (*p* < 0.05 shown by dark-colored filled squares. The Benjamini–Hochberg method was used to correct for multiple testing.

**Figure 8 ijms-23-08685-f008:**
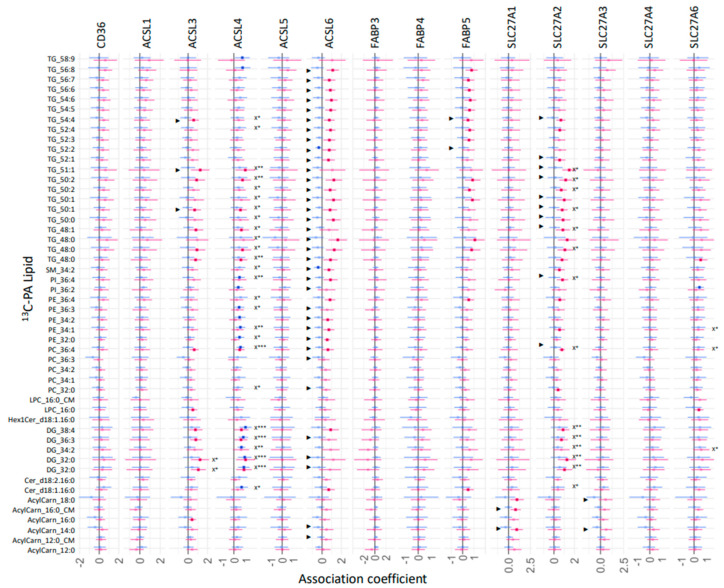
Sex-stratified associations between the amount of ^13^C-PA lipids (outcome) with relative mRNA abundance after 48 h of culture. Lipid amount was log2 transformed, mRNA abundance was log-2 transformed and z-scored. Linear regression with males and females combined (not shown) with significance *p* < 0.05 is indicated by X*, whilst significance *p* < 0.01 or 0.001 is indicated by X** and X***. Interaction between sex and maternal glycemia on lipid amount were then investigated with significant interactions *p* < 0.05 denoted by “►”. Linear regression was then run separately for males and females. Forest plots show association coefficient and 95% confidence intervals for males (blue) and females (red) with significance (*p* < 0.05 shown by dark-colored filled circles. The Benjamini–Hochberg method was used to correct for multiple testing.

**Figure 9 ijms-23-08685-f009:**
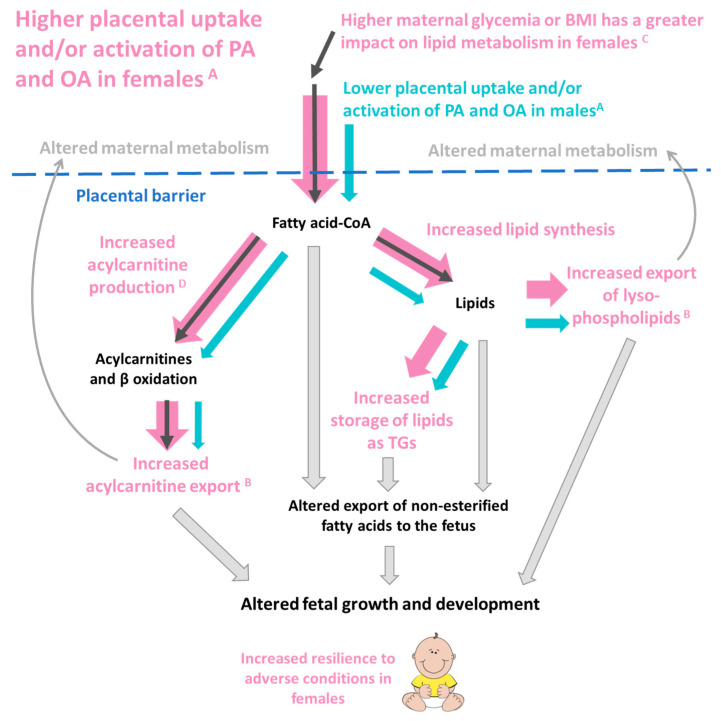
A postulation of how the sex-dependent placental metabolism of palmitic acid (PA) and oleic acid (OA) may influence offspring outcomes. ^A^ Female placenta produce more OA and PA labeled lipids from every measured lipid class, suggesting a greater capacity for fatty acid uptake and activation. ^B^ Female placenta also export a higher proportion of lysophospholipids and acylcarnitines compared to males. ^C^ In females, but not males, increases in maternal glycemia or BMI are associated with increases in placental PA and OA acylcarnitines and increases in PA lipids generally. In females, increased PA and OA acylcarnitines ^D^ are associated with increased birthweight. Thus, increased placental fatty acid uptake and incorporation in females (represented by the fatter pink arrows) could influence placental lipid reservoirs, local placental lipid activity, maternal–fetal lipid transfer and placental signaling to both fetus and mother. Placental lipid processing could influence maternal and fetal metabolism through both direct and indirect lipid-mediated pathways and hence impact fetal development.

**Table 1 ijms-23-08685-t001:** Clinical characteristics of study population.

Clinical Characteristics	Total	Male	Female
*n*	22	11	11
Maternal Age (years)	33.2 (2.7)	33.7 (2.4)	32.6 (2.9)
Chinese: Indian ethnicity	13:9	6:5	7:4
Maternal BMI in first trimester (kg/m^2^)	25.6 (5.0)	25.4 (5.4)	25.8 (4.7)
GDM ^#^ (%)	50%	45%	54%
Fasting glycemia (mmol/L) ^#^	4.5 (0.4)	4.4 (0.2)	4.6 (0.5)
2-h glycemia (mmol/L) ^#^	7.5 (1.6)	7.2 (1.8)	7.7 (1.5)
Gestational age at delivery (days)	271.2 (5.3)	270.5 (5.4)	271.9 (5.4)
Birthweight (g)	3215.7 (370.5)	3205 (423.3)	3226.4 (329.9)
Birthweight Centile (%) ^^^	57 (33.3)	54.9 (37.3)	59.2 (30.5)

Data presented as Mean (SD). ^#^ In mid-gestation 75 g three-time-point OGTT. Gestational diabetes (GDM) was defined by the WHO 2013 criteria: fasting plasma glucose 5.1–6.9 mmol/L or 1 h plasma glucose ≥ 10.0 mmol/L or 2 h PG 8.5–11.0 mmol/L. ^^^ Customized for gestational age using local references.

## Data Availability

The data that support the findings of this study are available on request from the corresponding author.

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
