# Peer review of "Sex-Dependent Regulation of Placental Oleic Acid and Palmitic Acid Metabolism by Maternal Glycemia and Associations with Birthweight"

_ijms, 2022, doi:10.3390/ijms23158685_

Round 1

Reviewer 1 Report

IJMS – Watkins et al. “Sex-dependent regulation of placental oleic acid and palmitic acid metabolism by maternal glycemia and associations with birthweight”

Using the placental explant model, Watkins et al. study the impact of maternal BMI, glycemia and fetal sex on placental metabolism and lipid synthesis of stable isotope labeled palmitic and oleic acids. This is very rich set of lipidomics data and the analysis has been thoughtfully approached. It is a large amount of data for a single manuscript and the discussion just scratches the surface of what could be explored here. That said, it is an important contribution to the literature. Detailed comments:

1.       1. The authors looked at lipid metabolism after 24 and 48 hours of explant exposure to tracers, but little was discussed about the differences between timepoints. What is the significance of these differences? Explant remodeling beginning after 24 hrs in culture may impact lipid incorporation and metabolism and some discussion of these factors would aid in interpretation.

2.     2.  It was found that female placenta have increased lipid synthesis, but no change in fatty acid transporter expression. In addition to the authors’ speculation that post-transcriptional regulation of transporters could be impacted by sex, their findings do not rule out changes in expression of metabolic genes such as CPT1, DGAT, FAS, etc.

3.       3. Given the amount of data and number of factors analyzed, a summary diagram that reflects the authors’ thoughts of how OA and PA metabolism (e.g. uptake, b-oxidation, storage and efflux) are impacted by maternal glycemia/BMI in males and females would be a valuable addition to the manuscript.

Reviewer 2 Report

In the manuscript authors assessed the impact of fetal sex on the lipid metabolism in cultured placental villous explants. The article is well-written and adequate methods were implemented. Certain information, however, need to be provided:

  1. The study population included women diagnosed with GDM. What type of treatment did the patients receive – only diet or diet with insulin? Insulin can modulate the expression of certain transporters, therefore, it would be advisable to provide such information and to discuss it’s potential impact.

Round 2

Reviewer 2 Report

I have no furthter comments.